# Enhanced Adipose Expression of Interferon Regulatory Factor (IRF)-5 Associates with the Signatures of Metabolic Inflammation in Diabetic Obese Patients

**DOI:** 10.3390/cells9030730

**Published:** 2020-03-16

**Authors:** Sardar Sindhu, Shihab Kochumon, Reeby Thomas, Abdullah Bennakhi, Fahd Al-Mulla, Rasheed Ahmad

**Affiliations:** 1Animal & Imaging Core Facility, Dasman Diabetes Institute (DDI), Al-Soor Street, P.O. Box 1180, Dasman 15462, Kuwait; 2Immunology & Microbiology, Dasman Diabetes Institute (DDI), Al-Soor Street, P.O. Box 1180, Dasman 15462, Kuwait; shihab.kochumon@dasmaninstitute.org (S.K.); reeby.thomas@dasmaninstitute.org (R.T.); 3Medical division, Dasman Diabetes Institute (DDI), Al-Soor Street, P.O. Box 1180, Dasman 15462, Kuwait; abdullah.bennakhi@dasmaninstitute.org; 4Genetics & Bioinformatics, Dasman Diabetes Institute (DDI), Al-Soor Street, P.O. Box 1180, Dasman 15462, Kuwait; fahd.almulla@dasmaninstitute.org

**Keywords:** Interferon regulatory factor-5, type-2 diabetes, obesity, adipose tissue, metabolic inflammation

## Abstract

Interferon regulatory factors (IRFs) are emerging as the metabolic transcriptional regulators in obesity/type-2 diabetes (T2D). IRF5 is implicated with macrophage polarization toward the inflammatory M1-phenotype, nonetheless, changes in the adipose expression of IRF5 in T2D and relationship of these changes with other markers of adipose inflammation remain unclear. Therefore, we determined the IRF5 gene expression in subcutaneous adipose tissue samples from 46 T2D patients including 35 obese (Body Mass Index/BMI 33.83 ± 0.42 kg/m^2^) and 11 lean/overweight individuals (BMI 27.55 ± 0.46 kg/m^2^) using real-time qRT-PCR. IRF5 protein expression was assessed using immunohistochemistry and confocal microscopy. Fasting plasma glucose, insulin, HbA1c, C-reactive protein, cholesterol, low- and high-density lipoproteins (LDL/HDL), and triglycerides were measured using commercial kits. IRF5 gene expression was compared with that of signature inflammatory markers and several clinico-metabolic indicators. The data (mean ± SEM) show the enhanced adipose IRF5 gene (*p* = 0.03) and protein (*p* = 0.05) expression in obese compared to lean/overweight diabetic patients. Adipose IRF5 transcripts in diabetic obese individuals associated positively with those of TNF-α, IL-18, IL-23A, CXCL8, CCL2, CCL7, CCR1/5, CD11c, CD68, CD86, TLR4/7/10, Dectin-1, FGL-2, MyD88, NF-κB, IRF3, and AML1 (*p* < 0.05). In diabetic lean/overweight subjects, IRF5 expression associated with BMI, body fat %age, glucose, insulin, homeostatic model assessment of insulin resistance (HOMA-IR, C-reactive protein (CRP), IL-5, and IL-1RL1 expression; while in all T2D patients, IRF5 expression correlated with that of IRF4, TLR2/8, and CD163. In conclusion, upregulated adipose tissue IRF5 expression in diabetic obese patients concurs with the inflammatory signatures and it may represent a potential marker for metabolic inflammation in obesity/T2D.

## 1. Introduction

In the pathogenesis of type-2 diabetes (T2D), obesity plays out as an independent risk factor of pivotal complications including insulin resistance and chronic low-grade inflammation called metabolic inflammation which is initiated by metabolic and inflammatory cells responding to overnutrition and a positive energy balance. The exact triggers of obesity-associated metabolic inflammation remain unclear and may also differ among tissues. Increasing evidence supports that metabolic inflammation is a consequence of obesity and it may play, in turn, a causative role in blunting insulin sensitivity and disrupting metabolic/energy homeostasis [1,2]. White adipose tissue (WAT) is known to store triglycerides from which lipids are mobilized to meet energy requirements as needed. WAT is subdivided into subcutaneous and visceral (abdominal/omental) depots with specific pathophysiological attributes and metabolic functions. Subcutaneous fat is the largest fat depot in humans which accounts for nearly 70–80% of total body fat, while visceral fat is the second largest fat depot which comprises about 10–15% of total body fat [3]. The changes occurring in the expanding WAT in obesity are considered central to the onset of metabolic inflammation and development of insulin resistance. Whereas, small adipocytes in lean individuals are linked to metabolic homeostasis, the enlarged adipocytes in obese individuals recruit activated macrophages as well as secrete adipokines [4]. Proinflammatory cytokines/chemokines secreted by resident macrophages and adipokines secreted by enlarged adipocytes act cooperatively to promote the adipose inflammation and lead to insulin resistance [5].

The WAT is now known to secrete proinflammatory cytokines such as TNF-α, IL-6, and IL-1β [6] as well as several chemokines such as CXCL8 (IL-8) and CCL2 (MCP-1) [7,8]. TNF-α is a signature proinflammatory cytokine which is secreted by both macrophages and adipocytes, especially by enlarged adipocytes in the visceral adipose tissue. TNF-α is involved in glucose and insulin metabolism, lipolysis, and insulin resistance [9]. Plasma TNF-α levels were found to be associated with visceral fat mass and body mass index (BMI) in patients with obesity/T2D [10]. Increased adipose tissue TNF-α expression in obesity was reported to be associated with insulin resistance [11], as well as with weight loss and lipoprotein lipase levels [12]. It was suggested that TNF-α-mediated adipocyte lipolysis due to lipoprotein lipase inhibition could lead to the increased circulatory levels of non-esterified fatty acids and insulin resistance [13]. Another proinflammatory mediator secreted predominantly by adipocytes is IL-6 which is also expressed by macrophages and other cells including T-lymphocytes, fibroblasts, adipose stromal cells, endothelial cells, and skeletal muscle [14]. Similar to TNF-α, IL-6 inhibits the expression of lipoprotein lipase but unlike TNF-α, it does not promote lipolysis. We and others have reported the increased adipose tissue IL-6 expression in obesity/T2D which could have consequences either as a circulating pleiotropic adipocytokine or as a local regulator of adipose inflammation and insulin action [15,16,17]. Other inflammatory mediators that can be elevated in obesity/T2D and promote metabolic inflammation include IL-18, IL-23, and IL-1β [18,19]. The chemotactic cytokines that direct leukocytic migration along concentration gradients are celled chemokines. The signature inflammatory chemokines that are implicated with metabolic inflammation belong to the subfamilies CXC (e.g., CXCL-1,5,8,9), CC (e.g., CCL-2,3,4,5,7,19), and CX3C (e.g., CX3CL-1). Notably, elevated levels of these chemokines in obesity/T2D have been linked to adipose tissue inflammation and insulin resistance [20,21,22].

The family of transcription factors called as interferon regulatory factors (IRFs) play a regulatory role in immunity and induction of type I interferons (IFN-α/β) [23]. IRFs constitute a family of 9 transcription factors named IRF-1 through IRF-9 that are implicated with immunoregulation and immunocyte differentiation via involvement of Toll-like and other pattern recognition receptors (PRR). The specific regions of IRFs bind to the corresponding motifs of their co-activators, co-repressors and modifiers to result in specific/diverse molecular events, based on cell types and stimuli involved. The emerging evidence supports the novel role of IRFs as transcriptional regulators of adipogenesis [24,25]. IRF5 orchestrates the macrophage polarization into inflammatory M1 phenotype and it positively modulates the adipose tissue deposition and insulin resistance in obesity [26]. We recently reported that increased adipose tissue IRF5 expression in non-diabetic obese individuals associated with BMI, percent body fat (PBF), and potential markers of adipose inflammation [20]; however, subcutaneous adipose tissue IRF5 expression in the diabetic obese patients and the relationship between IRF5 expression and signatures of metabolic inflammation still remain unclear. Herein, we present the data supporting that IRF5 expression in the subcutaneous adipose tissue associate with BMI, PBF, insulin, and homeostatic model assessment of insulin resistance (HOMA-IR) in diabetic lean/overweight patients while the adipose tissue IRF5 upregulation parallels with adipose inflammatory signatures in diabetic obese patients.

## 2. Materials and Methods

### 2.1. Subjects, Anthropometric Measurements, and Clinical Assays

A total of 46 T2D patients were recruited in the study through outpatient clinics of Dasman Diabetes Institute, Kuwait. T2D was defined as a fasting plasma glucose level of ≥7.0 mmol/L and use of anti-diabetic drugs [27]. Those with serious morbidities including pulmonary, renal, hepatic, cardiovascular, hematologic or immune diseases, type-1 diabetes, pregnancy, or malignancy were excluded. Based on BMI, study participants were divided into two groups as lean/overweight (<25/25–30 kg/m^2^) and obese (>30 kg/m^2^). The co-morbidities included 4 hypertension and 2 hyperlipidemia patients in diabetic lean/overweight group and 17 hypertension and 6 hyperlipidemia patients in diabetic obese group. Clinical characteristics of the study patients are summarized below in Table 1. All participants gave the written informed consent and the study was approved (RA# 2010-003/2015-027; June 2010/April 2016) by the ethics committee of Dasman Diabetes Institute, called ORA-DDI, which also includes non-affiliated members (physicians/scientists), a legal expert, an ethicist, and a community representative from non-scientific background. The ORA-DDI follows the updated guidelines and ethical principles for medical research involving human subjects as per the WMA declaration of Helsinki.

### 2.2. Anthropometry and Clinical Assays

Anthropometric and physical parameters of the study included body weight, height, waist circumference, and systolic/diastolic blood pressure. Body weight was measured using calibrated portable electronic weighing scale and height by using portable inflexible height measuring bars. Waist circumference at the highest point of the iliac crest and the mid-axillary line was measured using constant tension tape at the end of normal expiration and arms relaxed at sides. Whole body composition, i.e., PBF, soft lean mass and total body water were measured using IOI353 body composition analyzer (Jawon Medical, South Korea). Blood pressure was measured with Omron HEM-907XL digital automatic sphygmomanometer (Omron Healthcare Inc. IL, USA). Three blood pressure readings, with 5–10 min rest in between, were obtained. BMI was calculated as follows: BMI = Body weight (Kg)/Square of Height (m^2^). Peripheral blood was collected at phlebotomy unit from overnight-fasting (10h minimum) subjects. Plasma was collected by centrifuging heparinized blood (1200× *g* for 10 min at 4 °C) and serum was collected after blood was coagulated at room temperature for 30 min and later centrifuged at 1800× *g* for 10 min at 4 °C. Samples were aliquoted and stored at −80 °C until use. Fasting blood glucose, glycated hemoglobin (HbA1c), fasting serum insulin, and serum lipids were analyzed. Glucose and lipid profiles were detected using Siemens dimension RXL chemistry analyzer (Diamond Diagnostics, Holliston, MA, USA). HbA1c was measured using Variant™ device (BioRad, Hercules, CA, USA). Plasma triglycerides were measured using commercial kit (Intra-assay CV% = 0.93; Inter-assay CV% = 3.05) (Chema Diagnostica, Monsano, Italy). Levels of plasma C-reactive protein (CRP) and adiponectin were also measured by using commercial kits (Cat. No. DY1707 Human CRP DuoSet ELISA kit and Cat. No. DRP300 Human Total Adipokine/Acrp30 Quantikine ELISA kit, R&D systems, USA) and all assays were carried out as recommended by the manufacturers.

### 2.3. Collection of Adipose Tissue Biopsies

Human subcutaneous adipose tissue samples, ≈0.5 g each, were collected through abdominal subcutaneous fat-pad biopsy lateral to the umbilicus [15]. Briefly, periumbilical area was sterilized by swabbing with alcohol and anesthetized locally by injecting 2 mL of 2% lidocaine. Subcutaneous adipose tissue sample was collected through a small, 0.5 cm long skin incision. The collected fat sample was further incised, rinsed in cold PBS, fixed with 4% paraformaldehyde for 24 h and embedded in paraffin. At the same time, freshly collected fat samples, 50–100 mg each, were also preserved in RNAlater and stored at −80 °C until use.

### 2.4. Multiplex Quantitative Real-Time Reverse-Transcription Polymerase Chain Reaction (qRT-PCR)

Total RNA was purified from adipose tissue samples by using RNeasy kit and following the manufacturer’s instructions (Qiagen, Valencia, CA; USA) as described elsewhere [15]. The quantity of isolated RNA was determined by using Epoch™ Spectrophotometer System (BioTek, Winooski, VT, USA) and quality was assessed by formaldehyde-agarose gel electrophoresis. Samples (1 μg RNA each) were reverse transcribed into cDNAs using random hexamer primers and TaqMan reverse transcription reagents (High Capacity cDNA Reverse Transcription Kit; Applied Biosystems, CA, USA). The cDNA (50 ng) was amplified using TaqMan^®^ Gene Expression MasterMix (Applied Biosystems, CA, USA) and target gene-specific 20× TaqMan Gene Expression Assays (Applied Biosystems, CA, USA) containing forward/reverse primers of target genes including inflammatory cytokines/chemokines, chemokine receptors, inflammatory macrophage markers, innate immune TLR/non-TLR markers, TLR-associated signaling molecules and related transcription factors (Appendix A) and target-specific TaqMan^®^ minor groove binder (MGB) probes labeled with 6-fluorescein amidite (FAM) dye at 5’ end and non-fluorescent quencher (NFQ)-MGB at 3’ end of the probe, with 40 cycles of PCR amplification using 7500 Fast Real-Time PCR System (Applied Biosystems, CA, USA). Each thermal cycle included denaturation (95 °C, 15 s), annealing/extension (60 °C, 60 s), uracil DNA glycosylase activation (50 °C, 120 s), and AmpliTaq Gold enzyme activation (95 °C, 10 min). The amplified glyceraldehyde 3-phosphate dehydrogenase (GAPDH) gene expression was used as internal control to normalize individual sample differences. The expression level of each target gene relative to control (lean adipose tissue) was calculated by using 2^−ΔΔCt^ method and expressed as relative mRNA expression or fold change over the average control expression taken as 1. 

### 2.5. Immunohistochemistry (IHC)

For immunohistochemical staining, 4µm thick formalin-fixed paraffin-embedded adipose tissue sections were processed as described elsewhere [15]. Briefly, after antigen retrieval and blocking with 5% non-fat milk, slides were incubated overnight at room temperature with rabbit anti-human primary antibodies (Abcam, Cambridge, MA, USA) against IRF5 (ab140593, 1:400 diluted; Cambridge, MA, USA), TNF-α (ab9635, 1:800 diluted; Cambridge, MA, USA), CXCL8 (ab106350, 1:200 diluted; Cambridge, MA, USA), and CCL2 (ab9669, 1:400 diluted; Cambridge, MA, USA). Following three washings with PBS-Tween, samples were incubated (1h) with secondary antibody, i.e., goat anti-rabbit Alexa Fluor 594-conjugated antibody (ab150088, 1:200 diluted; Cambridge, MA, USA) and color was developed by using chromogenic substrate 3,3ʹ-diaminobenzidine (DAB; Cambridge, MA, USA). Specimens were washed, counterstained, dehydrated, cleared, and mounted as described elsewhere [28]. For analysis, digital photomicrographs (100× magnification; Scale bar 50 μm; Olympus BX51 Microscope, Japan) of adipose tissue were used to assess the staining intensity of 3 different regions marked by ImageScope software (Aperio Vista, CA, USA). Aperio-positive pixel count algorithm (version 9) was used to determine the staining intensity, expressed as arbitrary units (AU). The number of positive pixels was normalized to total number of pixels (positive and negative). The color and intensity thresholds were set to detect the immunostaining as positive and the background as negative pixels. The same parameters were used to analyze all samples and resulting color markup of analysis was confirmed for each slide.

### 2.6. Confocal Microscopy (CM)

Regarding CM, 8 µm thick formalin-fixed paraffin-embedded adipose tissue sections were processed using similar method as IHC. Following antigen retrieval and blocking, samples were incubated overnight at room temperature with primary antibodies (Abcam; Cambridge, MA, USA) as follows. In the panel showing IRF5 expression with adipocytes, samples were incubated with mouse anti-human IRF5 monoclonal antibody (ab140593, 1:50 diluted; Cambridge, MA, USA) and rabbit anti-human FABP4 polyclonal antibody (ab92501, 1:400 diluted; Cambridge, MA, USA). In the panel showing IRF5 expression with macrophages, samples were incubated with mouse anti-human IRF5 monoclonal antibody (ab140593, 1:50 diluted; Cambridge, MA, USA) and rabbit anti-human CD163 polyclonal antibody (ab87099, 1:400 diluted; Cambridge, MA, USA). After three washings with PBS-Tween, samples were incubated at room temperature for 1h with secondary antibodies as follows. Each group of slides, i.e., IRF5 staining with adipocytes and IRF5 staining with macrophage, was treated with goat anti-mouse Alexa Fluor 647-conjugated antibody (ab150115, 1:400 diluted, red fluorescence; Cambridge, MA, USA) and goat anti-rabbit Alexa Fluor 488-conjugated antibody (ab150077, 1:400 diluted, green fluorescence; Cambridge, MA, USA). Samples were washed thrice as before and finally counterstained with 4’,6-diamidino-2-phenylindole (DAPI; Vectashield, Vector Laboratories, H1500, blue fluorescence) and mounted. Confocal images (63× magnification; Scale bar 20 μm) were obtained by using inverted Zeiss LSM710 spectral confocal microscope (Carl Zeiss, Gottingen, Germany) and EC Plan-Neofluar 40×/1.30 oil DIC M27 objective lens. Following sample excitation with 543 nm HeNe laser and 405 nm line of argon ion laser, optimized emission detection bandwidths were configured by using Zeiss Zen 2010 control software.

### 2.7. Statistical Analysis

The data obtained were expressed as mean ± SEM values and group means were compared by using student’s t-test and associations between adipose IRF5 gene expression and other markers were determined using Pearson’s correlation coefficients (r). GraphPad Prism software (version 6.05; San Diego, CA, USA) was used for statistical analysis and for graphic presentation of the data. All P-values ≤0.05 were considered as statistically significant.

## 3. Results

### 3.1. Increased Adipose Tissue IRF5 Expression in Diabetic Obese Patients

WAT is a fat storage and endocrine organ which undergoes critical metabolic changes in obesity, insulin resistance and in the course of the development of T2D. We asked if the expression of metabolic transcription regulator IRF5 was differentially modulated in diabetic obese versus diabetic lean/overweight patients. To this effect, our data show that IRF5 mRNA expression was significantly higher in diabetic obese patients as compared to diabetic lean/overweight patients (*p* = 0.03) (Figure 1A). As expected, adipose IRF5 protein expression was also found to be elevated in diabetic obese patients compared to diabetic lean/overweight individuals (*p* = 0.05) (Figure 1B). The adipose IRF5 gene expression correlated positively with metabolic indicators of BMI (*r* = 0.62, *p* = 0.04) (Figure 1C) and PBF (*r* = 0.60, *p* = 0.05) (Figure 1D) only in diabetic lean/overweight group. Again in this group, IRF5 adipose gene expression was found to associate with plasma insulin (*r* = 0.77, *p* = 0.05), HOMA-IR (*r* = 0.76, *p* = 0.01) and CRP levels (*r* = 0.90, *p* = 0.004), whereas, no such associations were found regarding diabetic obese patients (see Table 2 below). Notably, adipose IRF5 gene and protein expressions were found to be mutually concordant (*r* = 0.50, *p* = 0.03) (Figure 1E).

The increase in the adipose IRF5 protein expression was also confirmed by CM. The representative IHC and CM images obtained from 3 independent determinations with similar results show IRF5 protein expression in the adipose tissue samples from diabetic lean, overweight and obese patients (Figure 2).

### 3.2. Elevated Adipose TNF-α Protein Expression in Diabetic Obese Patients Conforms with IRF5 Expression

Next, we wanted to know if IRF5 expression changes in the adipose tissue were consistent with the tissue expression of TNF-α which is a signature inflammatory adipo-cytokine involved in the pathophysiology of obesity/T2D. To this end, our data show that TNF-α protein expression was significantly elevated in diabetic obese individuals compared to diabetic lean/overweight subjects (*p* = 0.05) (Figure 3A), whereas the difference of TNF-α mRNA expression was non-significant (*p* = 0.09) (Figure 3B). A good agreement was found between the gene and protein expression of TNF-α in these individuals (*r* = 0.70, *p* = 0.0004) (Figure 3C). The representative IHC images from 3 independent determinations with similar results show TNF-α protein expression (arrows) in the adipose tissue samples from diabetic lean, overweight, and obese individuals (Figure 3D). Importantly, adipose TNF-α gene expression associated positively with adipose IRF5 gene expression in diabetic obese patients (*r* = 0.40, *p* = 0.02), but not in diabetic lean/overweight patients (*r* = 0.02, *p* = 0.96) (see Table 2 above).

### 3.3. CXCL8 Expression in Diabetic Obese Individuals Associates Positively with IRF5 Expression in the Adipose Tissue

CXCL8, also known as IL-8, is a critical α-chemokine produced mainly by macrophages and other cells which induces chemotaxis of neutrophils and other granulocytes. We asked if the adipose tissue expression of CXCL8 and IRF5 paralleled with each other in our diabetic patients. In this regard, the data show that adipose CXCL8 expression was elevated in diabetic obese compared to diabetic lean/overweight patients, both at protein (*p* = 0.02) (Figure 4A) and mRNA levels (*p* = 0.05) (Figure 4B). Furthermore, a direct association was found between the gene and protein expression of CXCL8 in the adipose tissue (*r* = 0.93, *p* < 0.0001) (Figure 4C). The representative IHC images from 3 independent determinations with similar results show the comparative CXCL8 protein expression (arrows) in the adipose tissue samples from diabetic lean, overweight, and obese patients (Figure 4D). Importantly, a positive correlation was found between the adipose gene expression of CXCL8 and IRF5 in diabetic obese patients (*r* = 0.41, *p* = 0.02) but not in lean/overweight subjects (*r* = 0.10, *p* = 0.74) (see Table 2 above).

### 3.4. Increased Adipose CCL2 Protein Expression in Diabetic Obese Patients

CCL2, also known as MCP-1, is a β-chemokine largely involved in the recruitment of monocytes, dendritic cells and memory T cells to the sites of inflammation. We wanted to know if CCL2 expression was associated with IRF5 gene expression in the adipose tissue. To this effect, our data show that adipose CCL2 protein expression was upregulated in diabetic obese patients compared to diabetic lean/overweight patients (*p* = 0.0003) (Figure 5A) whereas, CCL2 transcripts in two groups differed non-significantly (*p* = 0.10) (Figure 5B). Adipose CCL2 gene and protein expressions were found to be mutually concordant (*r* = 0.44, *p* = 0.05) (Figure 5C). The representative IHC images from 3 independent determinations with similar results show the comparative adipose CCL2 protein expression (arrows) in diabetic lean, overweight, and obese patients (Figure 5D). Of note, adipose gene expression of CCL2 associated positively with IRF5 gene expression in diabetic obese patients (*r* = 0.50, *p* = 0.004) but not in diabetic lean/overweight subjects (*r* = 0.01, *p* = 0.98) (see Table 2 above).

### 3.5. Relationship of IRF5 Gene Expression with Metabolic Markers and Inflammatory Signature in the Adipose Tissue

The adipose tissue IRF5 gene expression was found to be positively associated with clinico-metabolic indicators (BMI, PBF, insulin levels, and HOMA-IR) as well as inflammatory markers (TNF-α, CXCL8, CCL2, and CRP). As shown in Table 2 above, adipose tissue IRF5 transcripts expression in diabetic obese patients also correlated positively with the adipose gene expression of a wide variety of meta-inflammatory factors including: (i) proinflammatory cytokines, chemokines, and chemokine receptors e.g., IL-18 (*r* = 0.44, *p* = 0.01), IL-23A (*r* = 0.44, *p* = 0.008), CCL-7 (*r* = 0.40, *p* = 0.02), CCR1 (*r* = 0.51, *p* = 0.002), and CCR5 (*r* = 0.75, *p* < 0.0001); (ii) inflammatory M1 macrophage markers e.g., CD11c (*r* = 0.38, *p* = 0.02), CD68 (*r* = 0.63, *p* < 0.0001), CD86 (*r* = 0.61, *p* = 0.0002), and CD163 (*r* = 0.67, *p* < 0.0001); (iii) innate immune TLR/non-TLR proteins e.g., TLR2 (*r* = 0.78, *p* < 0.0001), TLR4 (*r* = 0.70, *p* < 0.0001), TLR7 (*r* = 0.50, *p* = 0.003), TLR8 (*r* = 0.73, *p* < 0.0001), TLR10 (*r* = 0.45, *p* = 0.008), Dectin-1 (*r* = 0.62, *p* < 0.0001), IL-1RL1 (*r* = 0.64, *p* = 0.02, only in lean/overweight group), and FGL-2 (*r* = 0.40, *p* = 0.02); (iv) TLR-downstream signaling molecules e.g., MyD88 (*r* = 0.64, *p* < 0.0001) and NF-κB (*r* = 0.50, *p* = 0.003); and (*v*) metabolic transcriptional regulators e.g., IRF3 (*r* = 0.40, *p* = 0.04), IRF4 (*r* = 0.36, *p* = 0.05), and AML1 (also called RUNX1) (*r* = 0.34, *p* = 0.05). Nonetheless, IL-5 is an anti-inflammatory/TH2 cytokine and IL-5 transcripts correlated negatively with adipose IRF5 mRNA expression only in diabetic lean/overweight patients (*r* = −0.60, *p* = 0.05). In essence, a thematic illustration presented below summarizes the afore-mentioned findings of this study (Figure 6).

## 4. Discussion

The WAT expansion in obesity has been linked to metabolic inflammation and also with an increased risk of developing insulin resistance, metabolic syndrome and T2D. IRFs are emerging as the key regulators of metabolic inflammation and their expression levels are expected to influence the immunobiological consequences in patients with obesity/T2D. Whereas, previous studies decipher the positive regulatory role of IRF3 and negative regulatory role of IRF4 in inflammation [29,30], changes in the adipose tissue IRF5 expression and their relationship with inflammatory signatures in this compartment remain elusive. Herein, we report for the first time to our knowledge, that the adipose tissue expression of metabolic transcriptional regulator IRF5 was enhanced significantly in the diabetic obese compared to diabetic lean/overweight patients. Further, the elevated IRF5 gene expression associated positively with metabolic indicators such as BMI and PBF only in diabetic lean/overweight but not in diabetic obese individuals. Notably, the changes in adipose IRF5 expression at transcriptional (mRNA) and translational (protein) levels were found to be mutually concordant. The transcriptional regulator IRF5 is viewed as an orchestral conductor of macrophage polarization from an anti-inflammatory M2 to an inflammatory M1 phenotype in the adipose tissue which links it directly with the nature and immunobiology of adipose tissue in obesity/T2D and related complications, In agreement with this argument, it was found that the knockdown of IRF5 in mouse model of high fat diet-induced obesity led to the preferential expansion of inguinal fat (equivalent to subcutaneous WAT in humans) than epididymal fat (equivalent to visceral WAT in humans) as compared to wild-type mice fed on high fat diet [26]. In the present study, increased adipose tissue IRF5 protein expression, as detected by both IHC and CM, was more remarkable in the diabetic obese than diabetic lean/overweight patients which suggests that obesity may play as a positive modulator of adipose IRF5 gene expression in diabetic patients; however, a direct association between adipose IRF5 gene expression and BMI/PBF was more explicit in the diabetic lean/overweight than diabetic obese individuals. It remains unclear whether more frequent medications and/or co-morbidities seen in the diabetic obese group could play out as confounding factors in this regard. Besides, IRF5 expression levels in the visceral WAT also remain unknown in these subjects. Since, in obesity major fat accumulation takes place in the abdominal cavity, i.e., a state of central obesity, IRF5 expression changes in the visceral fat may thus be more relevant to the parameters of BMI and PBF in the obese population. Interestingly, Dalmas et al. have reported higher visceral (epididymal in mice) than subcutaneous (inguinal in mice) adipose tissue expression of IRF5 in mice as well as humans. This study further elaborated that both mRNA and protein levels of IRF5 were higher in visceral WAT from morbidly obese individuals compared to non-obese subjects as well as higher IRF5 mRNA expression in visceral WAT from patients with metabolic syndrome compared to IRF5 mRNA expression (in decreasing order) in obese, overweight, and lean subjects [26] 

Also, our data show that in diabetic lean/overweight patients, but not in diabetic obese patients, adipose tissue IRF5 gene expression was associated positively with plasma insulin as well as HOMA-IR which is an alternative to the glucose clamp. HOMA-IR represents a surrogate measure of insulin resistance in vivo and is applicable to large epidemiologic applications. Although widely used, its cut-off to assess insulin resistance varies depending on multiple factors such as age, ethnicity, metabolic disease progression etc. Also, HOMA-IR may identify the insulin resistance phenotypes without directly measuring the insulin action. Not surprisingly, there is barely a consensus on cut-offs for tendencies toward insulin resistance or its classification. Different assays that are used for insulin detection may also influence the HOMA-IR values [31]. Given these factors, HOMA-IR variabilities may be envisaged in diabetic obese versus other categories of diabetic patients. In agreement with this argument, we found a positive association between adipose IRF5 gene expression and HOMA-IR only in the diabetic lean/overweight individuals but not in diabetic obese subjects. However, no previous data are available to verify these preliminary findings. Increased CRP levels are considered a clinical potential risk marker for metabolic inflammation and were found to be independently associated with T2D [27]. In this regard, we found that like HOMA-IR, systemic CRP levels were associated with adipose tissue IRF5 gene expression only in the diabetic lean/overweight patients but not in diabetic obese patients. We speculate that this may be due to more discernible patterns of systemic inflammation developing progressively in the overweight subjects than in obese individuals with T2D. 

As an active endocrine organ, WAT secretes a wide range of proinflammatory cytokines, chemokines and adipokines which play critical roles in the adipose inflammation to impair glucose metabolism and insulin sensitivity, leading eventually to the development of T2D [32]. As a fundamental and exacerbating causal factor of metabolic inflammation, activated circulatory monocytes respond to chemoattractants such as CCL2 and infiltrate into the expanding adipose tissue where they differentiate and colonize as the adipose tissue macrophages (ATMs). ATMs are the predominant source of proinflammatory cytokines and chemokines in WAT in obesity/T2D. Our data show significantly elevated adipose tissue expression of typical proinflammatory cytokines and chemokines including TNF-α, CXCL8, and CCL2 in diabetic obese compared to diabetic lean/overweight subjects. TNF-α is a critical proinflammatory cytokine and its overexpression in the adipose tissue has been reported in obese animals and humans [33,34]. In obesity/T2D, TNF-α is largely expressed in the adipose tissue by adipocytes, ATMs, monocytes, mast cells, lymphocytes, granulocytes, endothelial cells and fibroblasts. By inhibiting insulin-stimulated tyrosine phosphorylation of insulin receptor and insulin receptor substrate-1, TNF-α can play a crucial role in systemic insulin resistance and also decipher a clear link between obesity, insulin resistance and T2D [11,35,36]. CXCL8 and CCL2 represent the α- and β-chemokines, respectively, and may contribute to the adipose tissue inflammation via chemotaxis of inflammatory cells such as monocytes/macrophages, neutrophils and mast cells. CXCL8 is secreted by adipocytes, monocytes, macrophages, T-lymphocytes, and endothelial cells while CCL2 is secreted primarily by monocytes, macrophages and dendritic cells [37]. Of note, CXC chemokines are generally chemoattractant for neutrophils and CC chemokines mostly attract monocytes and lymphocytes. The both types of chemokines act as proinflammatory mediators and impair the insulin-stimulated glucose uptake over time and their elevated expression has been linked to obesity/T2D [38,39]. In our diabetic obese population, other inflammatory cytokines and chemokines that were found elevated in the adipose tissue included IL-18, IL-23A, and CCL7 (MCP-3). Regarding chemokine receptors, CCR1 and CCR5 were also highly expressed in the fat tissue samples from these individuals. Consistent with these findings, Zaharieva et al. have also reported the increased IL-18 expression in subjects with T2D and latent autoimmune diabetes of the adult while the IL-18 levels were correlated with lipid, glycemic and inflammatory parameters only in diabetic patients [40]. Our data showing upregulated adipose tissue expression of IL-23A are relatively new and not much is known how this cytokine influences the other players involved in metaflammation or insulin resistance in obese and/or T2D subjects. However, a previous study reported that IL-23 was implicated with β-cell ER stress, glucose intolerance and insulin resistance in obese mice [41]. Our data showing the elevated adipose tissue expression of other chemokine (CCL7) or chemokine receptors (CCR1 and CCR5) in diabetic obese patients are also corroborated, at least in part, by several other studies [42,43,44]. Our data indicating a direct association of IRF5 expression with those of inflammatory cytokines, chemokines, and chemokine receptors in the adipose tissue are consistent with other reports [45,46,47]. We also found that the anti-inflammatory proteins, such as IL-5 and IL-1RL1 (ST2) were expressed at high levels, especially in the adipose tissue samples from diabetic lean/overweight patients compared to diabetic obese patients. This differential expression of these two anti-inflammatory proteins between diabetic overweight and diabetic obese patients may have significance regarding the differences in their metabolic inflammatory status. In agreement with this argument, a previous study showed that the adipose tissue expression of IL-1RL1 and its ligand IL-33 played an anti-inflammatory role and protected mice from metabolic inflammation [48]. Similarly, IL-5 together with IL-10 and IFN-γ, were found to contribute to the maintenance of adipose tissue homeostasis in lean individuals [49].

In metaflammatory diseases, macrophage polarization and function undergo characteristic changes. In this regard, a typical M1-type proinflammatory macrophage polarization was observed in the adipose tissue in obesity and/or T2D which was found to be associated with hyperglycemia, hyperinsulinemia, increased TNF-α expression and adipose tissue hypoxia [50,51]. Consistent with these reports, we found that the expression of inflammatory macrophage markers including CD11c, CD68, and CD86 was elevated in the adipose tissue samples from diabetic obese patients compared to diabetic lean/overweight subjects which was also found to be associated positively with the adipose tissue expression of IRF5 transcripts. In adipose tissue from obese humans, proinflammatory ATMs were found to show high expression of CD11c, CD68, and CD86 as well as several other inflammation-associated markers viz CD36, CD40, CD64, HLA-DR, and TLR4 [52]. Interestingly, our confocal microscopy data reveal that IRF5 expression was mainly confined to macrophages that were found in the interstitial spaces around adipocytes, forming crown-like structures, in the obese adipose tissue while level of expression was relatively less in overweight and rather non-existent in lean adipose tissue. This observation is in agreement with previous reports indicating that the IRF5 expression was detected on macrophages in the fat (CD14^+^CD11c^+^/CD14^+^CD68^+^ cells) or in the liver [26,53]. Nonetheless, in addition to macrophages, IRF5 expression may also be found in other myeloid cells including neutrophils where it is linked with neutrophil function as well as with their role in acute inflammatory condition such as antigen-induced inflammatory arthritis and lung injury [54].

TLRs are the emerging key players in metabolic inflammation caused by chronic expression of proinflammatory cytokines, chemokines and adipokines in the adipose tissue in obesity/T2D. Therefore, we determined the relationship between adipose tissue expression of IRF5 and TLRs and their related receptors or downstream signaling molecules. To this end, our data show that elevated IRF5 expression was positively associated with the expression of TLR4, TLR7, TLR10, Dectin-1, and FGL-2 in diabetic obese patients. In line with this finding, increased expression of TLRs 1-9 and TLRs 11-13 was reported in two murine models of obesity [55]. Whereas in obese/T2D individuals, we have already reported the increased expression of several TLRs including TLR-2, -4, -7, -8, and -10 [28,56,57,58]. The present data show the congruence between adipose tissue expression of IRF5 and most TLRs in obesity/T2D. Similar to TLRs, adipose tissue expression of Dectin-1 was also found to be increased in diabetic obese patients which, as a non-TLR PRR, may have consequences for metabolic inflammation. Notably, an animal model study previously showed that Dectin-1 knock-out mice were protected from diet-induced obesity and insulin resistance and these mice also had reduced numbers of proinflammatory CD11c^+^ ATMs [59].

In the present study, increased IRF5 transcripts in the adipose tissue were found to be directly associated with those of MyD88, IRF3, NF-κB, and AML1. MyD88 is a proximal signaling canonical adaptor protein involved in inflammatory signaling pathways that are downstream of several TLRs and IL-1R, linking them to the IL-1R-associated kinase (IRAK) family kinases [60]. NF-κB is a critical transcription factor that regulates the cellular responses. Being a rapid-acting primary transcription factor, NF-κB acts as the first responder to cellular stress stimuli. Ligand binding and TLR stimulation, through downstream activation of IRAK family kinases, lead to the activation of NF-κB, mitogen-activated protein kinases (MAPKs) and activator protein-1 (AP-1). NF-κB is a major canonical transcription factor that regulates the inflammatory gene expression. Not surprisingly, NF-κB is chronically activated in several inflammatory conditions including obesity, T2D, sepsis, arthritis, inflammatory bowel disease, Crohn’s disease, asthma, and atherosclerosis [61,62,63,64,65]. In addition to NF-κB being the major driver of adipose inflammation, IRF3 may also act as a regulator of adipose inflammation. AML1 is a transcription factor that regulates differentiation of hematopoietic stem cells into mature blood cells as well as impacts beigeing or transition of white into beige fat i.e., a shift from the ‘fat storing’ to the ‘fat burning’ cells [66]. Consistent with our finding of the increased expression of these markers, at least in part, similar changes in the adipose tissue expression of MyD88, IRF3, NF-κB, and AML1 have been reported in obesity and/or T2D [28,29,66].

However, the present study involves a few caveats that warrant caution while interpreting these data. First off, the limited availability of fat biopsy samples did not allow us to separate the adipocytes and stromal cell fractions to analyze cell-specific changes within the WAT; nonetheless, the co-staining data suggest that IRF5 expression may be restricted mainly to immune cells such as macrophages. Second, limited subcutaneous fat volume availability also precluded the IHC and/or CM analyses for proinflammatory markers other than IRF5, TNF-α, CXCL8, and CCL2. Third, the data herein shown relate to the subcutaneous fat, hence, plausible changes in the visceral fat still remain unclear. Fourth, the stimuli inducing IRF5 expression and the functional significance of these changes like effects on insulin-stimulated glucose uptake in adipocytes and immunocytes also remain to be investigated. These concerns and other related aspects will be addressed in our future studies.

Taken together, our data show that the adipose tissue IRF5 gene and protein expressions were significantly elevated in diabetic obese patients. The increased IRF gene expression conformed with a wide range of inflammatory signatures in the adipose tissue including inflammatory cytokines/chemokines, chemokine receptors, macrophage markers, TLRs, as well as TLR-associated signaling molecules and transcription factors, implicating that IRF5 upregulation in the subcutaneous adipose tissue may represent a potential marker of metabolic inflammation in diabetic obese individuals.

## Figures and Tables

**Figure 1 cells-09-00730-f001:**
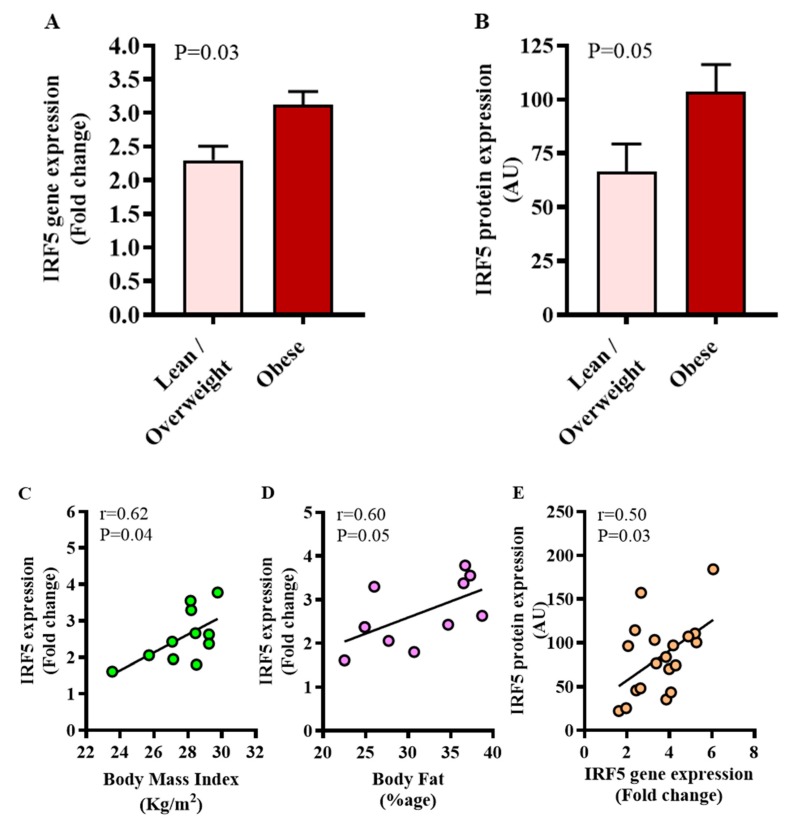
Increased adipose tissue interferon regulatory factor (IRF)5 expression in diabetic obese patients. IRF5 gene expression was assessed in the adipose tissue by using qRT-PCR in 46 type-2 diabetic (T2D) patients and expression of the IRF5 protein was determined by immunohistochemistry (IHC) in 19 T2D patients as described in materials and methods. Regarding qRT-PCR, GAPDH gene expression was used as internal control. The expression level of *IRF5* gene relative to control (lean adipose tissue) was calculated by using 2^−ΔΔCt^ method and expressed as relative mRNA expression or fold change over the average control expression taken as 1. Regarding IHC, IRF5 protein staining intensity, expressed as arbitrary units (AU), was determined by using Aperio-positive pixel count algorithm and ImageScope software. The number of positive pixels was normalized to total pixels (positive and negative) and color/intensity thresholds were set with immunostaining as positive and background as negative pixels. The data (mean±SEM) show significantly elevated (**A**) IRF5 gene expression (fold change) (*p* = 0.03) and (**B**) IRF5 protein expression (AU) (*p* = 0.05) in diabetic obese compared to diabetic lean/overweight patients. Furthermore, in diabetic lean/overweight subjects (11), IRF5 gene expression was found to associate positively with (**C**) body mass index (BMI: *r* = 0.62, *p* = 0.04) and (**D**) %age of body fat (*r* = 0.60, *p* = 0.05). (**E**) Overall, IRF5 gene and protein expression were found to be mutually concordant (*r* = 0.50, *p* = 0.03).

**Figure 2 cells-09-00730-f002:**
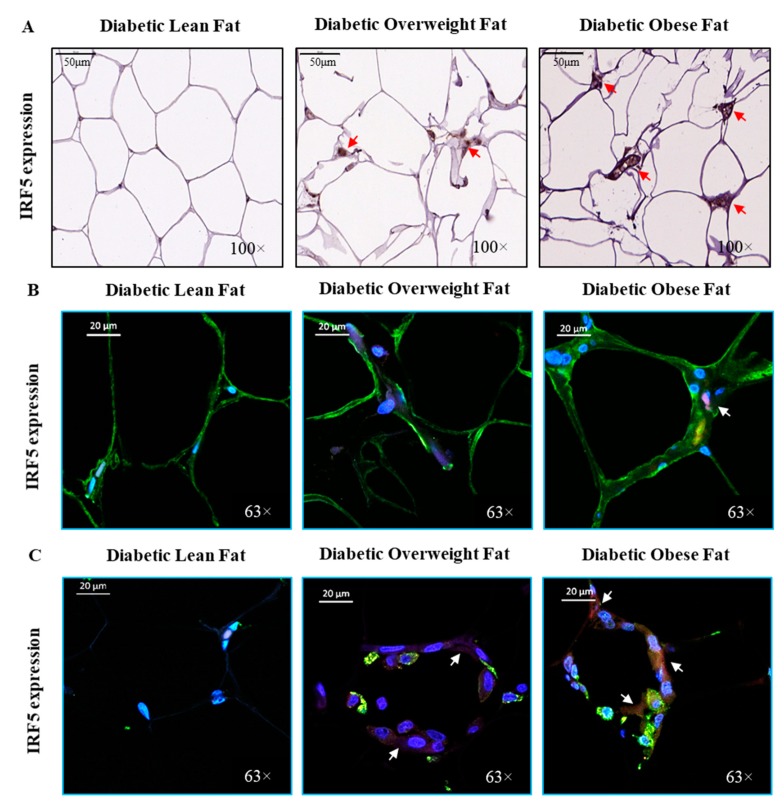
Comparative adipose tissue IRF5 expression in diabetic patients. Adipose IRF5 protein expression was determined by immunohistochemistry (IHC). IRF5 expression was also determined in adipocytes and macrophages using confocal microscopy (CM) as described in materials and methods. (**A**) The representative IHC images (100× magnification; scale bar 50 μm) obtained from three independent determinations, each, with similar results show the comparative adipose IRF5 protein expression (arrows) in diabetic lean, overweight, and obese patients. Similarly, representative CM images (63× magnification; scale bar 20 μm) obtained from three independent determinations with similar results show IRF5 expression (IRF5 staining in red; see arrows) in (**B**) adipocytes (FABP4 staining in green) and (**C**) macrophages (CD163 staining in green) for diabetic lean, overweight, and obese patients; while the blue color represents DAPI counterstaining.

**Figure 3 cells-09-00730-f003:**
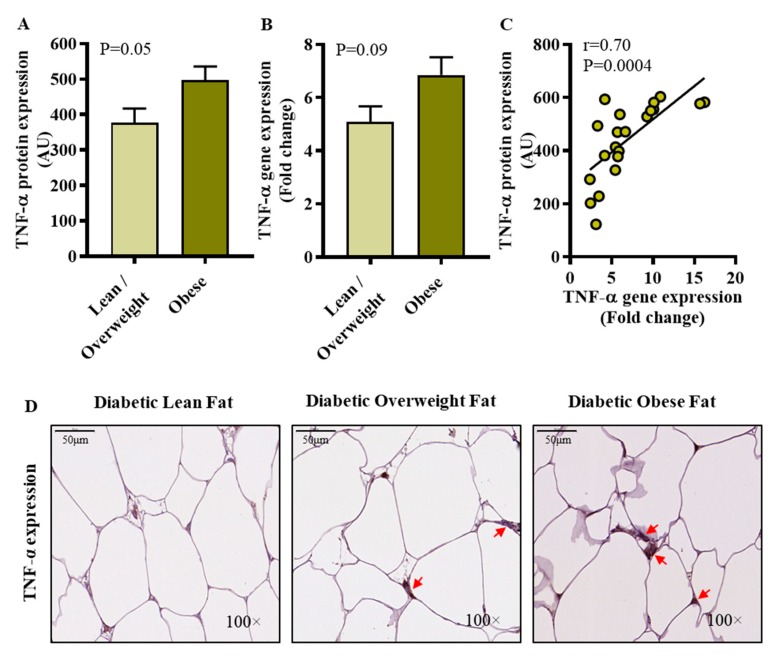
Elevated adipose TNF-α protein expression in diabetic obese patients. Adipose TNF-α protein and gene expressions were assessed by immunohistochemistry (IHC) and qRT-PCR in 21 and 46 type-2 diabetic patients, respectively, as described in materials and methods. Regarding IHC, TNF-α protein staining intensity expressed as arbitrary units (AU) was determined by using Aperio-positive pixel count algorithm and ImageScope software. The number of positive pixels was normalized to total pixels (positive and negative) and color/intensity thresholds were set with immunostaining as positive and background as negative pixels. Regarding qRT-PCR, GAPDH gene expression was used as internal control. The expression level of *TNF-α* gene relative to control (lean adipose tissue) was calculated by using 2^−ΔΔCt^ method and expressed as relative mRNA expression or fold change over the average control expression taken as 1. (**A**) The data (mean±SEM) show that adipose TNF-α protein expression (AU) was significantly higher in diabetic obese compared to diabetic lean/overweight patients (*p* = 0.05); (**B**) However, TNF-α transcripts’ expression (fold change) differed non-significantly between diabetic obese and diabetic lean/overweight patients (*p* = 0.09). (**C**) A positive association was found between TNF-α gene and protein expression (*r* = 0.70, *p* = 0.0004). (**D**) The representative IHC images from three independent determinations with similar results show the comparative TNF-α protein expression (arrows) in the fat tissue from diabetic lean, overweight, and obese patients (100× magnification; scale bar 50 μm).

**Figure 4 cells-09-00730-f004:**
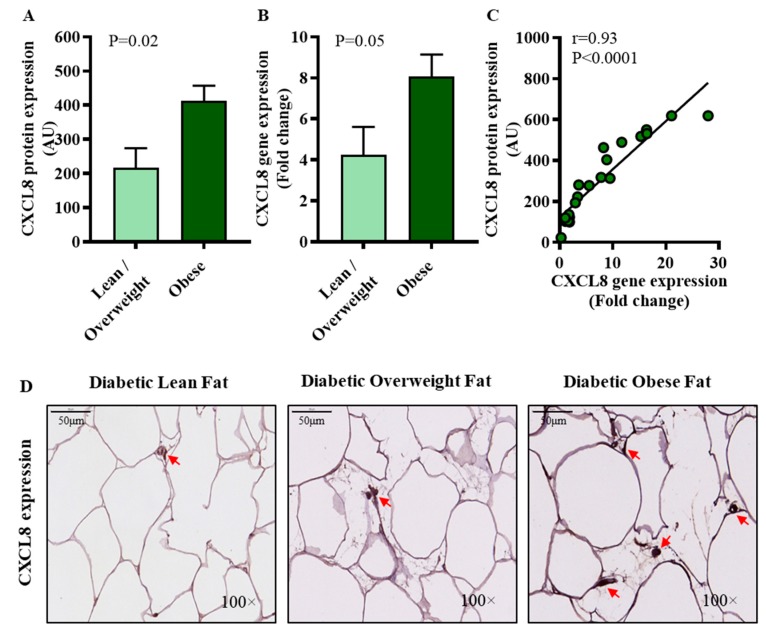
Increased adipose CXCL8 expression in diabetic obese patients. Adipose CXCL8 protein and gene expressions were assessed by immunohistochemistry (IHC) and qRT-PCR in 21 and 46 type-2 diabetic patients, respectively, as described in materials and methods. Regarding IHC, CXCL8 protein staining intensity expressed as arbitrary units (AU) was determined by using Aperio-positive pixel count algorithm and ImageScope software. The number of positive pixels was normalized to total pixels (positive and negative) and color/intensity thresholds were set with immunostaining as positive and background as negative pixels. Regarding qRT-PCR, GAPDH gene expression was used as internal control. The expression level of *CXCL8* gene relative to control (lean adipose tissue) was calculated by using 2^−ΔΔCt^ method and expressed as relative mRNA expression or fold change over the average control expression taken as 1. (**A**) The data (mean±SEM) show that adipose CXCL8 protein expression (AU) was significantly higher in diabetic obese patients compared to diabetic lean/overweight patients (*p* = 0.02). (**B**) As expected, adipose CXCL8 mRNA expression (fold change) was also higher in diabetic obese compared to diabetic lean/overweight patients (*p* = 0.05). (**C**) A strong positive correlation was found between the gene and protein expression of CXCL8 (*r* = 0.93, *p* < 0.0001). (**D**) The representative IHC images from three independent determinations with similar results show the comparative adipose CXCL8 protein expression (arrows) in diabetic lean, overweight, and obese patients (100× magnification; scale bar 50 μm).

**Figure 5 cells-09-00730-f005:**
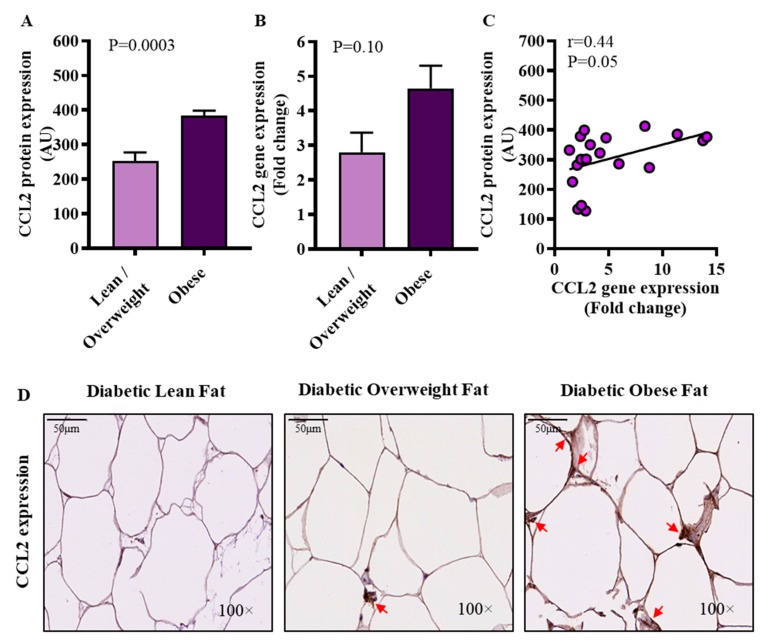
Increased adipose CCL2 protein expression in diabetic obese patients. Adipose CCL2 protein and gene expressions were determined by immunohistochemistry (IHC) and qRT-PCR in 21 and 46 type-2 diabetic patients, respectively, as described in materials and methods. Regarding IHC, CCL2 protein staining intensity expressed as arbitrary units (AU) was determined by using Aperio-positive pixel count algorithm and ImageScope software. The number of positive pixels was normalized to total pixels (positive and negative) and color/intensity thresholds were set with immunostaining as positive and background as negative pixels. Regarding qRT-PCR, GAPDH gene expression was used as internal control. The expression level of *CCL2* gene relative to control (lean adipose tissue) was calculated by using 2^−ΔΔCt^ method and expressed as relative mRNA expression or fold change over the average control expression taken as 1. (**A**) The data (mean ± SEM) show that CCL2 protein expression (AU) was significantly higher in diabetic obese patients compared to diabetic lean/overweight patients (*p* = 0.0003). (**B**) However, CCL2 transcripts (fold change) differed non-significantly between diabetic obese and diabetic lean/overweight patients (*p* = 0.10). (**C**) CCL2 gene and protein expressions correlated positively (*r* = 0.44, *p* = 0.05). (**D**) The representative IHC images from three independent determinations with similar results show the comparative adipose CCL2 protein expression (arrows) in diabetic lean, overweight, and obese patients (100× magnification; scale bar 50 μm).

**Figure 6 cells-09-00730-f006:**
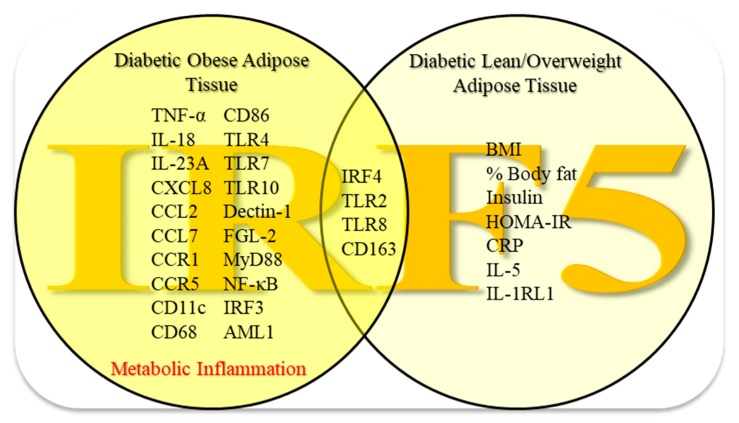
Enhanced adipose tissue IRF5 expression in diabetic obese patients parallels with the signatures of metabolic inflammation. The data presented in this study support a model of metabolic inflammation in type-2 diabetes, in which, increased adipose IRF5 gene expression in diabetic obese patients concurs with the local expression of a wide variety of inflammatory markers including TNF-α, IL-18, IL-23A, CXCL8, CCL2/7, CCR1/5, CD11c, CD68, CD86, TLR4/7/10, Dectin-1, FGL-2, MyD88, NF-κB, IRF3, and AML1. On the other hand, in diabetic lean/overweight patients, adipose IRF5 gene expression was found to correlate with BMI, body fat percentage, insulin levels, HOMA-IR index, plasma CRP, and adipose IL-5 and IL-1RL1 transcripts expression. In all diabetic patients, regardless of the status of their obesity, adipose IRF5 transcripts correlated with IRF4, TLR2/8, and CD163 gene expression. Altogether, these changes imply that the adipose tissue IRF5 upregulation may represent a novel marker of metabolic inflammation in type-2 diabetes.

**Table 1 cells-09-00730-t001:** Patients’ characteristics and clinical data.

Parameter	Diabetic Lean/Overweight	Diabetic Obese
Total number (N)	11 (7 male, 4 female)	35 (19 male, 16 female)
Age (Yrs.)	53.43 ± 1.15	52.34 ± 1.70
Body mass index (BMI) (kg/m^2^)	27.55 ± 0.46	33.83 ± 0.42
Body fat (%)	31.88 ± 1.61	37.59 ± 0.97
Fasting plasma glucose (mmol/L)	8.28 ± 0.67	8.74 ± 0.50
Fasting plasma insulin (mIU/L)	24.63 ± 8.13	25.68 ± 4.37
Homeostatic model assessment of insulin resistance (HOMA-IR) (Glucose×Insulin/22.5)	5.47 ± 1.82	9.25 ± 1.49
Glycated hemoglobin (HbA1c) (%)	7.53 ± 0.51	8.24 ± 0.24
C-reactive protein (CRP) (pg/mL)	6.41 ± 1.19	5.46 ± 0.78
Total cholesterol (mmol/L)	4.80 ± 0.48	4.99 ± 0.20
HDL (mmol/L)	1.11 ± 0.10	1.18 ± 0.05
LDL (mmol/L)	2.89 ± 0.40	2.93 ± 0.19
Triglycerides (mmol/L)	1.76 ± 0.25	1.84 ± 0.24
Hypertension (N)	4	17
Hyperlipidemia (N)	2	6
Therapy	Metformin, Lipitor, Diamicron, Zocor, NovoRapid, Concor, Insulin, Aldomet, Lantus, Diovan	Metformin, Lipitor, Diamicron, Lantus, NovoRapid, Concor, Insulin, Aldomet, Tenormin, Zestril

**Table 2 cells-09-00730-t002:** Correlation of the adipose IRF5 gene expression with various markers.

Marker Type	Diabetic Lean/Overweight	Diabetic Obese
Clinical/metabolic markers	BMI: *r* = 0.62, *p* = 0.04* PBF: *r* = 0.60, *p* = 0.05* Insulin: *r* = 0.76, *p* = 0.05* HOMA-IR: *r* = 0.77, *p* = 0.006** CRP: *r* = 0.95, *p* = 0.004** HbA1c: *r* = 0.01, *p* = 0.98 FBG: *r* = 0.14, *p* = 0.65 Cholesterol: *r* = 0.02, *p* = 0.96 HDL: *r* = 0.21, *p* = 0.48 LDL: *r* = 0.001, *p* = 0.99 Triglycerides: *r* = 0.27, *p* = 0.38	BMI: *r* = 0.10, *p* = 0.64 PBF: *r* = 0.07, *p* = 0.74 Insulin: *r* = 0.09, *p* = 0.67 HOMA-IR: *r* = 0.10, *p* = 0.65 CRP: *r* = 0.20, *p* = 0.32 HbA1c: *r* = 0.04, *p* = 0.81 FBG: *r* = 0.05, *p* = 0.79 Cholesterol: *r* = 0.25, *p* = 0.14 HDL: *r* = 0.10, *p* = 0.65 LDL: *r* = 0.25, *p* = 0.17 Triglycerides: *r* = 0.12, *p* = 0.49
Inflammatory cytokines/chemokines or chemokine receptors	TNF-α: *r* = 0.02, *p* = 0.96 IL-1β: *r* = 0.14, *p* = 0.66 IL-5: *r* = -0.60, *p* = 0.05* IL-6: *r* = 0.19, *p* = 0.54 IL-18: *r* = 0.01, *p* = 0.97 IL-23A: *r* = 0.10, *p* = 0.79 CXCL8: *r* = 0.10, *p* = 0.74 CXCL9: *r* = 0.06, *p* = 0.84 CXCL10: *r* = 0.14, *p* = 0.65 CCL2: *r* = 0.01, *p* = 0.98 CCL5: *r* = 0.10, *p* = 0.79 CCL7: *r* = 0.10, *p* = 0.75 CCL11: *r* = 0.07, *p* = 0.80 CCL19: *r* = 0.30, *p* = 0.32 CCR1: *r* = 0.14, *p* = 0.62 CCR2: *r* = 0.52, *p* = 0.06 CCR5: *r* = 0.34, *p* = 0.26	TNF-α: *r* = 0.40, *p* = 0.02* IL-1β: *r* = 0.18, *p* = 0.39 IL-5: *r* = −0.10, *p* = 0.71 IL-6: *r* = 0.10, *p* = 0.67 IL-18: *r* = 0.44, *p* = 0.01* IL-23A: *r* = 0.44, *p* = 0.008** CXCL8: *r* = 0.41, *p* = 0.02* CXCL9: *r* = 0.01, *p* = 0.95 CXCL10: *r* = 0.07, *p* = 0.70 CCL2: *r* = 0.50, *p* = 0.004** CCL5: *r* = 0.28, *p* = 0.14 CCL7: *r* = 0.40, *p* = 0.02* CCL11: *r* = 0.22, *p* = 0.21 CCL19: *r* = 0.12, *p* = 0.49 CCR1: *r* = 0.51, *p* = 0.002** CCR2: *r* = 0.10, *p* = 0.75 CCR5: *r* = 0.75, *p* < 0.0001****
Monocyte/macrophage markers	CD11c: *r* = 0.06, *p* = 0.85 CD68: *r* = 0.10, *p* = 0.74 CD86: *r* = 0.27, *p* = 0.36 CD163: *r* = 0.64, *p* = 0.01* CD302: *r* = –0.45, *p* = 0.10	CD11c: *r* = 0.38, *p* = 0.02* CD68: *r* = 0.63, *p* < 0.0001**** CD86: *r* = 0.61, *p* = 0.0002*** CD163: *r* = 0.67, *p* < 0.0001**** CD302: *r* = 0.30, *p* = 0.08
TLR/non-TLR innate immune markers	TLR2: *r* = 0.59, *p* = 0.03* TLR3: *r* = 0.12, *p* = 0.69 TLR4: *r* = 0.08, *p* = 0.79 TLR7: *r* = 0.43, *p* = 0.13 TLR8: *r* = 0.60, *p* = 0.04* TLR9: *r* = 0.32, *p* = 0.27 TLR10: *r* = 0.03, *p* = 0.91 Dectin-1: *r* = 0.10, *p* = 0.79 IL-1RL1: *r* = 0.64, *p* = 0.02* FGL-2: *r* = 0.14, *p* = 0.63	TLR2: *r* = 0.78, *p* < 0.0001**** TLR3: *r* = 0.27, *p* = 0.15 TLR4: *r* = 0.70, *p* < 0.0001**** TLR7: *r* = 0.50, *p* = 0.003** TLR8: *r* = 0.73, *p* < 0.0001**** TLR9: *r* = 0.22, *p* = 0.21 TLR10: *r* = 0.45, *p* = 0.008** Dectin-1: *r* = 0.62, *p* < 0.0001**** IL-1RL1: *r* = 0.22, *p* = 0.21 FGL-2: *r* = 0.40, *p* = 0.02*
TLR-associated signaling molecules and transcription factors	MyD88: *r* = 0.35, *p* = 0.24 NF-κB: *r* = 0.02, *p* = 0.95 IRF3: *r* = 0.26, *p* = 0.39 IRF4: *r* = 0.92, *p* < 0.0001**** AML1: *r* = 0.45, *p* = 0.10	MyD88: *r* = 0.64, *p* < 0.0001**** NF-κB: *r* = 0.50, *p* = 0.003** IRF3: *r* = 0.40, *p* = 0.04* IRF4: *r* = 0.36, *p* = 0.05* AML1: *r* = 0.34, *p* = 0.05*

Note: Number of asterisks corresponds to the statistical significance level.

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
