# Peer review of "Enhanced Adipose Expression of Interferon Regulatory Factor (IRF)-5 Associates with the Signatures of Metabolic Inflammation in Diabetic Obese Patients"

_cells, 2020, doi:10.3390/cells9030730_

Round 1

Reviewer 1 Report

In this manuscript Sindhu et al describe a close correlation between IRF5 and inflammatory cytokine expression in adipose tissue of obese, but not lean or overweight people with diabetes. IRF5 expression was also correlated with BMI, age, insulin, among other parameters. Tissues were analyzed using RT-qPCR and immunohistochemistry.

The data presented supports the authors’ hypothesis that IRF5 is associated with an inflammatory signature in obese diabetic patients. However to draw a firm conclusion, random gene sets (including equal number of genes) should be correlated with IRF5 expression to determine if this is a specific correlation. The authors also show immunohistochemistry to verify mRNA expression data, it would be of great value if these stainings would include immune cell and adipose marker genes to determine which cell type has enhanced expression.

Specific comments:

RT-PCR results are expressed as fold change, shouldn’t that be also arbitrary numbers or was a specific value set to 1? This should be clarified in all figures.

Figure 2, please indicate which proteins are labeled in the specific colors and add scale bars to the pictures. This also needs to be added to the methods section.

Author Response

Response to Reviewer 1 Comments

Point 1: In General Comments: Minor spell check required; & Design/Methods/Conclusions can be improved

Response 1: The whole text has been carefully checked for misspells, errors/omissions and any grammatical mistakes in the use of English language. Moreover, Design, Methods, and Conclusions sections have been modified as kindly suggested. Please see the revised manuscript with all changes marked.

Point 2: Under Comments and Suggestions for Authors: The data presented supports the authors’ hypothesis that IRF5 is associated with an inflammatory signature in obese diabetic patients. However, to draw a firm conclusion, random gene sets (including equal number of genes) should be correlated with IRF5 expression to determine if this is a specific correlation. The authors also show immunohistochemistry to verify mRNA expression data, it would be of great value if these stainings would include immune cell and adipose marker genes to determine which cell type has enhanced expression.

Response 2: Regarding IRF5 correlative associations with signature inflammatory markers, their co-expression was detected in the same adipose tissue sample from each patient by multiplex q-RT-PCR, using GAPDH as house-keeping gene/internal control to normalize the individual sample differences. Also, in compliance to the reviewer’s kind suggestion, new confocal microscopy images have been added to show IRF5 expression in adipocytes (co-staining marker FABP4) (Fig 2B) and IRF5 expression in macrophages (co-staining marker CD163) (Fig 2C) in the adipose tissue.

Point 3: RT-PCR results are expressed as fold change; shouldn’t that be also arbitrary numbers or was a specific value set to 1? This should be clarified in all figures.

Response 3: Relative mRNA expression i.e. expression level of each target gene relative to the control (lean adipose tissue) was calculated by using 2-ΔΔCt method and expressed as “fold change” over the average control expression taken as 1. This clarification has been accordingly added in all Figures 1, 3, 4 & 5.

Point 4: Figure 2, please indicate which proteins are labeled in the specific colors and add scale bars to the pictures. This also needs to be added to the methods section.

Response 4: New confocal microscopy images are now added in Fig 2 showing IRF staining (in red) in adipocytes (FABP4 marker stained as green) (Fig 2B) and IRF staining in macrophages (CD163 marker stained as green) (Fig 2C); while DAPI counter-staining is blue. Also, scale bars have been added to all images. IHC images (100× magnification) have scale bar 50μm and CM images (63× magnification) have scale bar 20μm. The relevant information is accordingly added to methods section under confocal microscopy.

Reviewer 2 Report

The manuscript by Sindhu et al. is an extension of published work by Dalmas and colleagues showing that IRF5 expression in adipose tissue from obese patients/donors is increased. One difference in the current study is the examination of IRF5 expression, along with cytokines/chemokines and innate immune signaling molecules, in lean and obese diabetic patients. While this does expand our current knowledge of IRF5 expression changes in obese versus obese diabetic patients, it was limited in identifying what cell type IRF5 expression is increased in, why IRF5 expression is increased, and what the functional contribution of increased IRF5 expression is. Thus, the findings are incremental. Correlations with inflammatory markers suggests that it might be leading to increased expression of these, yet this was not examined. Additional comments are listed below.

Please include primary references for the role of IRF3 (Kumari et al. 2016) and IRF4 (Eguchi et al. 2013) in adipose inflammation, as they are relevant to the current study and mentioned in the Discussion. Given the identified correlation between IRF5, IRF3 and IRF4 expression in Table 2, do the authors think there are redundant or overlapping roles between these IRFs? In the IHC and confocal experiments, it is unclear exactly how the authors are quantifying protein expression. Without the use of additional surface markers to identify in which cell type IRF5 is being expressed in the adipose tissue, it is difficult to make conclusions on adipose expression. This is particularly relevant as Dalmas et al. (2015) found that expression was actually coming from macrophages within the adipose tissue. Dalmas et al. found that IRF5 was regulating TGFb in their human adipose tissue, was TGFb expression analyzed in the current study from diabetic patients? Do the authors think their are distinctions between their model and findings from Dalmas et al. in the patient samples? The authors bring up the limitations of their study in the Discussion as to why they were unable to separate cells within the adipose tissue, but they could have performed co-staining with surface markers to identify the cell type expressing IRF5. This should be done and would greatly strengthen their findings.    

Author Response

Response to Reviewer 2 Comments

Point 1: In General Comments: Minor spell check required; as well as Background, Design, Methods & Conclusions need to be improved

Response 1: The whole text has been checked carefully for misspells, errors/omissions and any grammatical issues in the standard use of English language. As well, the Design, Methods and Conclusions sections have been modified as kindly suggested. Please see the revised manuscript with all changes marked.

Point 2: The manuscript by Sindhu et al. is an extension of published work by Dalmas and colleagues showing that IRF5 expression in adipose tissue from obese patients/donors is increased. One difference in the current study is the examination of IRF5 expression, along with cytokines/chemokines and innate immune signaling molecules, in lean and obese diabetic patients. While this does expand our current knowledge of IRF5 expression changes in obese versus obese diabetic patients, it was limited in identifying what cell type IRF5 expression is increased in, why IRF5 expression is increased, and what the functional contribution of increased IRF5 expression is. Thus, the findings are incremental. Correlations with inflammatory markers suggests that it might be leading to increased expression of these, yet this was not examined. Additional comments are listed below.

Response 2: These comments are important and are genuinely appreciated. As mentioned in MS text, this preliminary study is limited by certain caveats such as the concerns raised herein. With regard to which cell types show IRF5 expression, new co-staining data (confocal images) have been added, separately depicting the IRF5 expression (red) in adipocytes (FABP4 stained as green) (Fig 2B) and IRF5 expression (red) in macrophages (CD163 stained as green) (Fig 2C); and counter-staining by DAPI is blue. In agreement with previous study by Dalmas et al. IRF5 expression was confined to macrophages in the obese adipose tissue and this has been mentioned accordingly in the discussion section as well (Lines 540-548). Regarding the question as to why IRF5 expression is increased in obesity, our preliminary findings (un-published data) suggest that a fundamental trigger relevant to obesity i.e. oxidative stress (through ROS elevation) can induce the expression of potent proinflammatory factors such as TNF-α, IL-6, MCP-1 (CCL2) and IL-8 (CXCL8) in the human monocytic cells or macrophages. Of these factors, especially IL-8 can directly induce the IRF5 expression in these immune cells; however, these data need to be further validated using different cell models. Notably, IL-8 plasma concentrations have been reported to be increased in obese subjects which were found to correlate with BMI, body fat %age, fat mass and TNF-α system (sTNFR2/sTNFR1) (Ref. Straczkowski M. et al. Plasma interleukin-8 concentrations are increased … J Clin Endocrinol Metab 2002; 87: 4602-6).

Point 3: Please include primary references for the role of IRF3 (Kumari et al. 2016) and IRF4 (Eguchi et al. 2013) in adipose inflammation, as they are relevant to the current study and mentioned in the Discussion. Given the identified correlation between IRF5, IRF3 and IRF4 expression in Table 2, do the authors think there are redundant or overlapping roles between these IRFs?

Response 3: The relevant references have been added to Discussion part accordingly (Lines 427-429). In view of the published data, roles of IRF3 and IRF5 support an inflammatory phenotype of the adipose tissue in obesity/T2D and might be overlapping whereas IRF4 acts as a negative regulator of inflammation in diet induced obesity (Ref. Eguchi J. Interferon regulatory factor 4 regulates … Diabetes 2013; 62: 3394-403). In our study, a positive association of IRF5 with IRF3 and IRF4 alike, may imply that a counter-regulatory (homeostatic) mechanism also exists in presence of adipose tissue inflammation in obese diabetic individuals.

Point 4: In the IHC and confocal experiments, it is unclear exactly how the authors are quantifying protein expression.

Response 4: The protein expression was quantified in case of IHC images only. Aperio-positive pixel count algorithm (version 9) was used to determine the staining intensity expressed as arbitrary units (AU) in 3 different regions which were marked by ImageScope software (Aperio Vista, CA, USA) for each adipose tissue sample. The confocal microscopy images have been presented as such, with IRF5 positive staining indicated by white arrows.

Point 5: Without the use of additional surface markers to identify in which cell type IRF5 is being expressed in the adipose tissue, it is difficult to make conclusions on adipose expression. This is particularly relevant as Dalmas et al. (2015) found that expression was actually coming from macrophages within the adipose tissue. Dalmas et al. found that IRF5 was regulating TGFb in their human adipose tissue, was TGFb expression analyzed in the current study from diabetic patients? Do the authors think there are distinctions between their model and findings from Dalmas et al. in the patient samples? The authors bring up the limitations of their study in the Discussion as to why they were unable to separate cells within the adipose tissue, but they could have performed co-staining with surface markers to identify the cell type expressing IRF5. This should be done and would greatly strengthen their findings.

Response 5: The study by Dalmas et al. has reported increased IRF5 mRNA and protein expression in epididymal (visceral) fat than inguinal (subcutaneous) fat in diet-induced obese mice. The authors further reported that IRF5 mRNA expression was higher in visceral WAT from metabolic syndrome patients compared with obese (which had higher expression than overweight and lean subjects), thus concluding that IRF5 positively correlated with visceral adiposity in humans. As of TGF-β expression in our study participants, we determined TGF-β mRNA expression which did not correlate with IRF5 transcripts expression in diabetic obese (N=33, r=0.16, P=0.38) as well as diabetic lean/overweight subjects (N=11, r=0.50, P=0.10). Our study model deciphers a positive (correlative) link between IRF5 and the global inflammatory state of adipose tissue in diabetic obese individuals, based on expression of a wide range of inflammatory cytokines/chemokines, chemokine receptors, inflammatory macrophage markers, innate TLRs, related signaling molecule and transcription factors. Similarly, Dalmas et al. also found a positive correlation of IRF5 with ITGAX (encoding CD11c) & TLR4, as well as 32 other gene involved in JAK-STAT pathway. However, we did not find a correlation between IRF5 and TGF-β expression in the subcutaneous adipose tissue whereas Dalmas study found a negative association between IRF5 and TGF-β expression in the visceral adipose tissue. This discrepancy could possibly relate to immunometabolic differences between subcutaneous and visceral depots of white adipose tissue in obese/T2D patients. Regarding co-staining of IRF5 with surface markers for adipocytes (FABP4) and macrophages (CD163), new confocal images have been now added (Figs 2B and 2C), showing that IRF5 expression is mainly restricted to macrophages in the adipose tissue which is consistent with the study by Dalmas et al.

Reviewer 3 Report

In their publication entitled “enhanced adipose expression interferon regulatory factor 5 associates with the signatures of metabolic inflammation in diabetic obese patients” Sindhu et al have analysed the expression of IRF5 and other markers of inflammation in subcutaneous adipose tissue biopsies from patients with T2D, being obese or lean/overweight. Expression was correlated to serum and plasma parameters. They present data that supports that IRF5 expression correlates to expression of other inflammatory markers and with clinical markers of T2D (HOMA-IR, CRP, body fat%).

General comments:

This study is largely descriptive in nature and is limited in its contribution to the field. It has long been established that IRF5 is an important inflammatory marker in macrophages, in diabetes and in other inflammatory diseases (Saliba et al 2014 Cell Rep; Dalmas et al 2015 Nature Medicine). Although results support conclusions, these conclusions are not very original.  

- In fact, this paper seems to be a side-product of the group’s previous publication in November 2019 “Increased adipose tissue expression of IRF5 in obesity: association with metabolic inflammation”. The paper in 2019 is from the same institute with the same first and last authors, this is the definition of ‘salami slicing’ publication.  

- In fact, some written parts of the current paper under review are identical to that published in Nov 2019.  

For example:

o First paragraph of introduction  

o Results sections 3.2

Nov 2019: “3.2. IRF5 Gene Expression Correlates with that of TNF-α but not IL-6 in Adipose Tissue”

January 2020: “3.2. Elevated adipose TNF-α protein expression in diabetic obese patients conforms with IRF5 expression”

o Results sections 3.3  

Nov 2019: “3.3. Adipose CXCL8 Expression is Enhanced in Obese Individuals and Associates with IRF5 Expression”

January 2020: “CXCL8 expression in diabetic obese individuals associates positively with IRF5 expression in the adipose tissue”  

o Results sections 3.5  

Nov 2019: 3.5. Relationship of IRF5 Gene Expression with Signature Inflammatory Immune Markers in the Adipose Tissue

January 2020: 3.5. Relationship of IRF5 gene expression with metabolic markers and inflammatory signature in the adipose tissue

Author Response

Response to Reviewer 3 Comments

Point 1: This study is largely descriptive in nature and is limited in its contribution to the field. It has long been established that IRF5 is an important inflammatory marker in macrophages, in diabetes and in other inflammatory diseases (Saliba et al 2014 Cell Rep; Dalmas et al 2015 Nature Medicine). Although results support conclusions, these conclusions are not very original.  

Response 1: Regarding mouse model study by Saliba et al., LPS-stimulated macrophages were used to identify the IRF5:RelA interaction by protein:DNA microarray analysis. A direct role of IRF5:RelA cistrome in the regulation of specific inflammatory genes was demonstrated via global gene expression analysis of mouse macrophages deficient in IRF5 and RelA. Given that IRF5 recruitment to inflammatory gene promoters is aided by RelA, the authors suggested that interfering with RelA:IRF5 interaction could downregulate the inflammatory genes in macrophages. In other study by Dalmas et al. IRF5 was implicated in inflammatory macrophage polarization (M1-type) and insulin resistance in mice fed on a high-fat diet. The authors also showed that IRF5 expression in visceral adipose tissue was higher in patients with metabolic syndrome, followed in decreasing order by obese, overweight and lean subjects. Also, IRF5 was found to inhibit TGF-β expression in macrophages in the visceral adipose tissue of obese individuals. Given the afore-mentioned studies, it seems important to investigate IRF5 expression changes together with other inflammatory markers in the subcutaneous fat in diabetic obese vs. overweight/lean humans. The present study deciphers associations of IRF5 with a wide variety of inflammatory factors including proinflammatory cytokines, chemokines, chemokine receptors, M1 macrophage markers, TLRs, non-TLR innate factors, TLR-downstream signaling proteins and transcription factors. Thus, the present data add to what is previously known about IRF5 as a regulator of metabolic inflammation.

Point 2: In fact, this paper seems to be a side-product of the group’s previous publication in November 2019 “Increased adipose tissue expression of IRF5 in obesity: association with metabolic inflammation”. The paper in 2019 is from the same institute with the same first and last authors, this is the definition of ‘salami slicing’ publication. - In fact, some written parts of the current paper under review are identical to that published in Nov 2019. For example: o First paragraph of introduction; o Results sections 3.2: Nov 2019: “3.2. IRF5 Gene Expression Correlates with that of TNF-α but not IL-6 in Adipose Tissue”. January 2020: “3.2. Elevated adipose TNF-α protein expression in diabetic obese patients conforms with IRF5 expression”; o Results sections 3.3: Nov 2019: “3.3. Adipose CXCL8 Expression is Enhanced in Obese Individuals and Associates with IRF5 Expression”. January 2020: “CXCL8 expression in diabetic obese individuals associates positively with IRF5 expression in the adipose tissue”; o Results sections 3.5: Nov 2019: 3.5. Relationship of IRF5 Gene Expression with Signature Inflammatory Immune Markers in the Adipose Tissue. January 2020: 3.5. Relationship of IRF5 gene expression with metabolic markers and inflammatory signature in the adipose tissue

Response 2: We excuse for a repetitive expression of the text mentioned above. In fact, the current MS focuses specifically on diabetic obese patients; distinctly apart from non-diabetic obese that are reported previously. These data indicate that there are differences regarding IRF5 associations with a wide range of inflammatory factors in the subcutaneous adipose tissues from obese, overweight, and lean subjects. This is why diabetic patients’ data have been presented as a separate manuscript. For instance, in non-diabetics (previous publication), IRF5 adipose expression was inversely associated with local adiponectin expression while in diabetics (present manuscript), we did not find an association between IRF5 and adiponectin expression. IRF5 expression was found to be positively associated with HbA1c levels in non-diabetics (previous MS), but no such association was found in diabetics (present MS). IRF5 associated with TNF-α/CXCL-8 in both non-diabetic obese and overweight; however, this association was found only in diabetic obese and not in diabetic overweight/lean (present MS). Besides, IRF5 associations differed between diabetics and non-diabetics regarding a large number of inflammatory markers such as IL-1β, IL-23A, CCL2, CCL5, CXCL8, CXCL9, CXCL10, CCR2, TLR4, TLR9, TLR10, IRAK-1, NF-κB, Dextin-1, and FGL-2. That said, we respectfully state that presenting the diabetic patients’ data as a separate MS seems to be merited.

Round 2

Reviewer 2 Report

The manuscript has been greatly strengthened. Confocal images could use additional positive and negative controls, especially since the IRF5 antibody from Abcam is not longer available due to quality/specificity issues.